# Automated device for continuous stirring while sampling in liquid chromatography systems

Omer Markovitch [1,2✉], Jim Ottelé [2], Obe Veldman[3] & Sijbren Otto [2]

Ultra-performance liquid chromatography is a common analysis tool, and stirring is common in many laboratory setups. Here we show a device which enables continuous stirring of samples whilst inside an ultra-performance liquid chromatography system. Utilizing standard magnetic stirring bars that fit standard vials, the device allows for the automation of experimental setups that require stirring. The device is designed such that it can replace the standard sample holder and fits in its place, while being battery operated. The use of three-dimensional (3D) printing and commercially available parts enables low-effort and low-cost device production, as well as easy modifications. Testing the device was performed by video analysis and by following the kinetics of a dynamic combinatorial library that is known to be exquisitely sensitive to agitation, as a result of involving a fiber growth-breakage mechanism. Design files and schematics are provided.

[1] Origins Center, Groningen, The Netherlands. [2] Center for Systems Chemistry, Stratingh Institute, University of Groningen, Groningen, The Netherlands. [3] Veldman Technische Ontwikkeling en Advisering, Groningen, The Netherlands. ✉email: omermar@gmail.com

High- and ultra- performance liquid chromatography (HPLC and UPLC, respectively) are common analytical tools for the detection and identification of components in complex mixtures[1]. At the heart of these techniques is sampling and then separation of the sample by the use of a column with appropriate properties, leading to different compounds eluting at different times from the column.

Stirring a sample is fundamental in many experimental setups as it promotes mixture homogeneity. Furthermore, in systems where large assemblies are formed, mechanical agitation can lead to their breakage which may affect systems' behavior[2,3]. Stirring can be achieved by placing a (Teflon coated) magnetic bar within a sample and placing the sample over a device with a rotating magnetic field.

Here, we have developed a UPLC stirring device that replaces a standard sample holder and enables battery-powered magnetic stirring in a similar manner to standard laboratory stirring devices. An experimental setup that requires continuous stirring can now be run inside the UPLC instrument, allowing for multiple measurements at various times without the need for sample preparation or human intervention. We demonstrate the applicability of the device through the usage of a system that has previously been shown to be highly sensitive to stirring—exhibiting exponential growth that is enabled through a fiber elongation/breakage mechanism[4,5].

The development of this device employed open-source content and 3D printing[6–8], aligned with the increased do-it-yourself movement in science[9–12].

## Results and discussion

**Design of device.** The design and development of the device employed open source and readily available parts in order to make it readily available for reproduction. The device's dimensions are such that it is interchangeable with the manufacturer-provided sample holder (Fig. 1). Each vial holders' dimension and position exactly match the standard. A motor is used for rotating an internal plate with magnets such that the magnetic stirring bars inside the sample vials will rotate and thus stirring is achieved. The control module is separated from the stirring module. The stirring speed range is 200–1200 rpm (in steps of 100) and is controlled by an onboard microcontroller and an organic light-emitting diode screen. Control over stirring speed of the plate with magnets is achieved via an internal feedback of the actual measured speed. A display reports the battery status and rotation rate.

Any measured speed difference of more than 10% compared to the user set speed which lasts more than 60 s, is considered an error and is indicated by an exclamation mark on the screen, while the device continues to operate. When turning the device on, if there were errors detected in the last run, a summary of their duration will appear on-screen (grouped into differences of 10–20, 20–30, 30–40, and more than 40%).

**Validation.** To validate the new device, a dynamic system of dithiols was studied (Fig. 2) This system is an excellent candidate to validate the device as: (i) the material is heterogeneous and settles to the bottom when not properly stirred, and (ii) the kinetics are highly sensitive to shear stress, as it has previously been shown that when it is subjected to mechanical agitation hexamer macrocycles assemble into fibers and exponentially grow

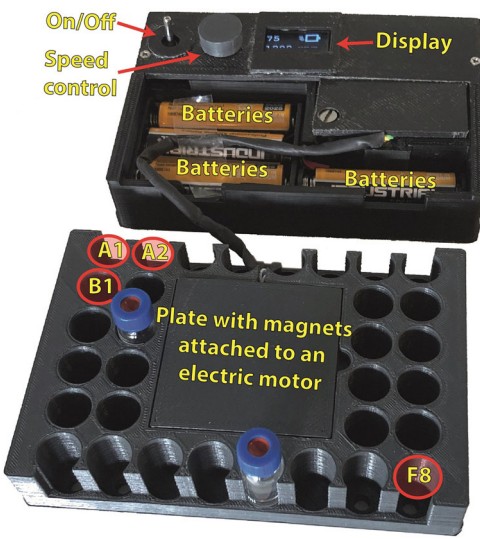

**Fig. 1 Device overview.** Sample holder positions are labeled A1 to F8.

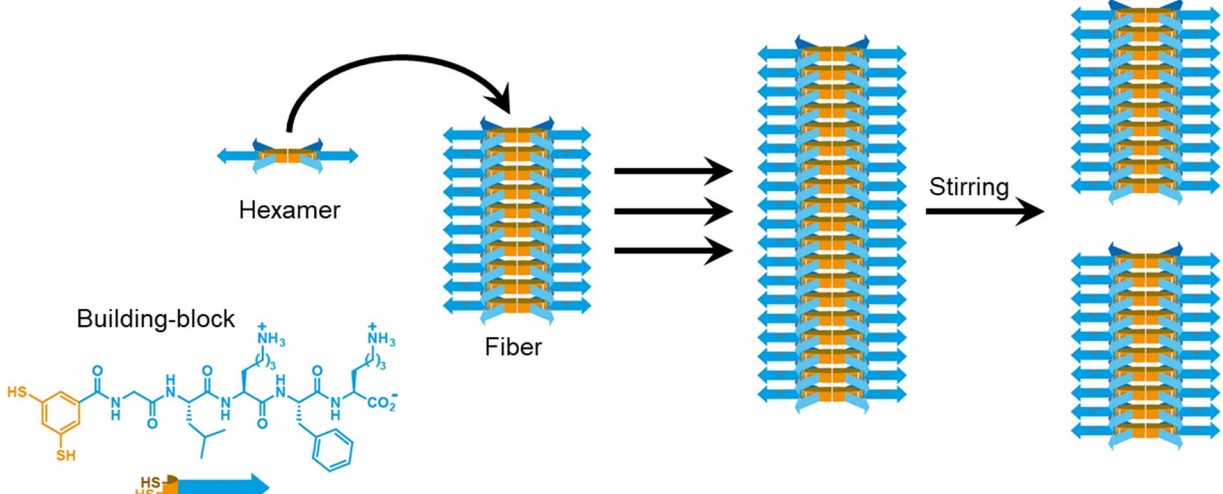

**Fig. 2 Chemical system used to probe reproducibility of agitation.** Oxidation of the dithiol building-block (either by air or by sodium perborate ($NaBO_3$)) leads to the formation of a mixture of disulfide macrocycles of different ring sizes (not shown). Of these, the hexamer self-assembles. The kinetics of autocatalytic formation of hexamer is highly sensitive to agitation, with growth rate depending linearly on the number of fiber ends.[5] Fibers are fragile and the number of fiber ends is determined by the agitation regime.

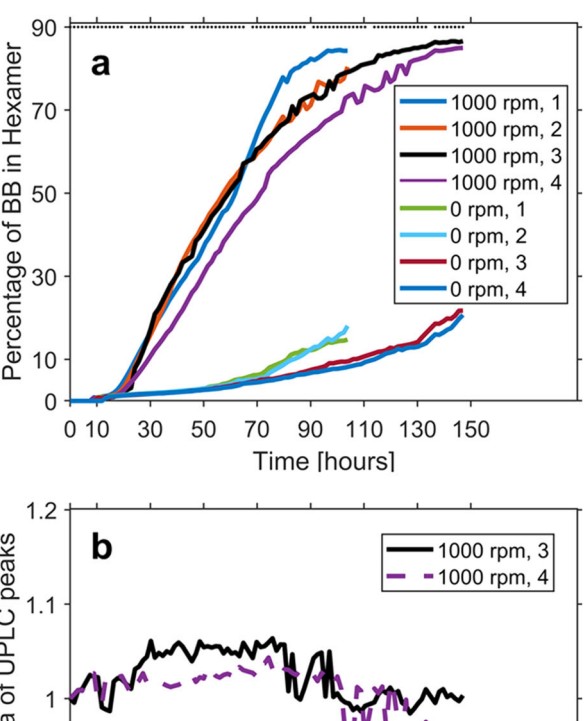

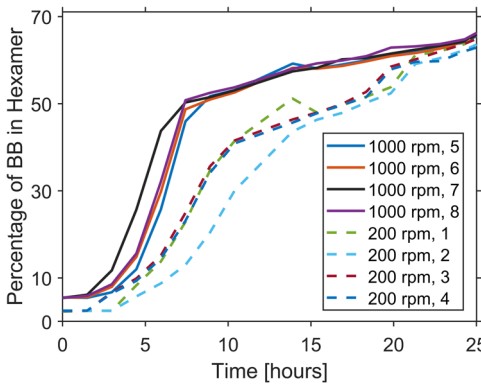

**Fig. 4 Hexamer emergence over time, under pre-oxidation with NaBO₃.** Time to reach 50% of building-blocks in hexamer: 8.17 ± 0.86 h (1000 rpm), 17.2 ± 2.2 h (200 rpm). Four repeats were conducted for each condition. Two separate sets of experiments were run, stirred at 1000 and 200 rpm respectively. Figure data are available in Supplementary Dataset 1.

**Fig. 3 Kinetics. a** Hexamer emergence over time, given in percentage out of the initial building-block (BB) concentration, for stirred and unstirred samples. Two separate sets of experiments were run, each with two stirred samples and two unstirred samples. Black dots in panel a indicate the time point where a measurement was taken. For each condition, four repeats were performed. **b** Total area of all UPLC peaks detected by the UPLC software, relative to the initial total area (respectively $1.752 \times 10^7$ and $1.849 \times 10^7$ [arbitrary units]). Figure data are available in Supplementary Dataset 1.

and undergo self-replication[4]. Initial formation of fiber seeds (nucleation) is a stochastic process, and exponential growth occurs through a fiber elongation-breakage mechanism[5] (Fig. 2). Due to the system's unique response to stirring, it is chosen to validate the device's performance.

Figure 3a shows the emergence kinetics of hexamers using the new stirring device, with and without stirring (4 repeats were performed for each case). Indeed, after approximately 100 h the total percentage of building-block mass in hexamers in the stirred samples reached an average value of 79 ± 5 vs only 13 ± 5 when unstirred, demonstrating the device's ability to consistently stir the samples through the course of the experiments. The small differences within each set of repeats is attributed to the stochastic nature of fiber nucleation. High throughput automatic sampling of the device allows for 60 to 90 consecutive measurements per each sample over the course of a week, and is indicated with small black dots on the very top of Fig. 3a. Animation of UPLC traces and a plot with chromatograms are respectively given in Supplementary Video 1 and Supplementary Note 3.

Given the high number of consecutive sampling it is important to test if the stirring quality is consistent throughout the experiments. If a sample is not homogeneously stirred then fibers precipitate and therefore, when the UPLC needle is sampling, the fibers are under-represented, which results in loss of peak area. Figure 3b shows that, at each time point, the total area of all peaks is well conserved throughout, with a relative standard deviation of only 2.27 and 2.30%, respectively.

Figure 4 shows that, indeed stirring at different speeds leads to different rates of exponential growth, reflecting the established role of fibers and their breakage in this process[5]. This is in agreement with previous observations. It is noted that at the elevated temperature at which experiments were performed here (40 °C), hexamer formation and replication speed are similar to the speed of building-block oxidation by air, which hampers observing differences under different stirring rates. Therefore, experiments shown in Fig. 4 were performed by first pre-oxidizing the building-block solution with NaBO₃ and then comparing 1000 with 200 rpm stirring rates (see Methods section).

In order to allow for a practical quantitative validation that does not depend on a specific chemical system, a video analysis was performed whereby the actual rotation speed of the Teflon coated magnet was measured in positions C2, D2, E2, F2, F3, A4, F4, A5, F5, C7 and D7 (Fig. 1), at a speed of 1000 and 200 rpm. These positions are chosen because they immediately surround the central motor, while positions in columns 1 and 8 exhibit diminished reproducibility of the stirring effect as they are further away from the device's center and experience a weaker magnetic field. This analysis finds an average speed of rotation which is 0.99 and 0.98 relative to the device's set rpm speed, thus confirming the device's functionality, i.e. only about 1% difference with the set stir speed (Supplementary Note 4 and Supplementary Videos 2). Due to symmetry, the video analysis also informs on positions A2, A3, A6, A7, B2, B7, E7, F7 and F8.

Starting from fresh batteries, the battery level is typically reduced to 35% after ~72 h of continuous stirring, at which point they were replaced.

**Conclusions**. A device for the continuous stirring of samples whilst inside a UPLC system is presented and validated. Such a device allows for the automation of experiments that require stirring and facilitates time-resolved UPLC measurements (e.g. for kinetics studies). The latter may be particularly advantageous when studying and modeling complex dynamic chemistries[13,14]. It is our belief that such a device can be beneficial for other analytical laboratories around the globe.

In principle it is possible to amend the present design to fit to other chromatography machines and extend its capabilities. It is also possible to program the onboard microcontroller to allow for more complicated scenarios.

## Methods

**Design and fabrication of the stirring device.** 3D printing was used for maximum versatility in design and manufacturing, as well as to allow user-specific modifications. User requests and stirring readouts are handled via ARDUINO Pro Mini 3.3 V 8 MHz microcontroller[15]. A standard motor is used for rotating a plate with magnets (magnet dimensions $25 \times 8 \times 1 \text{ mm}^3$). Control over precise rotation rate of the plate is achieved via a magnetic sensor that provides feedback to the microcontroller. See Supplementary Note 5 and Supplementary Note 6 for components details and instructions on how to reproduce the device. Supplementary Dataset 3 contains the Arduino's firmware code.

3D components have been printed using Creatr Duel Extruder (nuzzle diameter 0.35 mm) by Leapfrog and da Vinci 1.0 Pro (nuzzle diameter 0.40 mm) by XYZ Printers, using PLA 1.75 mm filament under the following settings, unless stated otherwise: printing temperature was 205 °C, print bed temperature was 40 °C and at low printing speed (for reference, print time of the main body of the sample holder was about 13 h, and printing time of the main body of the control unit was about 10 h). Components' design is given in STL format (Supplementary Dataset 2).

The device's profile is approximately 4 mm higher than the standard default Waters acquity UPLC sample holder (catalogue number 700005209), and consequently UPLC needle height was adjusted. An overview of the device is given in Fig. 1.

**Reaction setup.** A 2.0 mM aqueous stock solution was prepared by dissolving 1.2 mg of a dithiol building-block (Fig. 2. Peptide amino acids sequence: Gly–Leu–Lys–Phe–Lys) in 607 µL borate buffer (50 mM in boron atoms, pH 8.12). Samples were prepared by adding 250 µL of the stock solution to a UPLC vial (dimensions $12 \times 32$ mm) and diluting it with 750 µL borate buffer. A Teflon coated stirring bar (dimensions $5 \times 2$ mm) was added to some of the samples and the vials were closed with a Teflon-lined screw cap.

**Experiments and measurements.** After preparation, samples were placed within the stirring device inside the UPLC instrument, with stirring turned on. The system was monitored by subjecting it to periodic UPLC analysis using a Waters Acquity UPLC-H Class system equipped with a photo diode array detector. All analyses were performed using a reversed-phase UPLC column (Aeris Peptide 1.7 µm XB-C18x 2.10 mm, Phenomenex). The column temperature was kept at 35 °C, and the sample chamber was kept at 40 °C. UV absorbance was monitored at 254 nm. Additional occasional column washes were performed. The eluents used in the UPLC separation consist of $H_2O$ and acetonitrile, both are UPLC grade and contain 0.1 vol% trifluoroacetic acid. Gradient and peak integration algorithm are respectively given in Supplementary Note 1 and Supplementary Note 2.

For pre-oxidation experiments, the samples were prepared as described before, with the addition of 0.50 equivalents $NaBO_3$ so that 50% of the building-block amount is oxidized. Then, the resulting mixture was kept inside the UPLC sample holder and subjected to periodic UPLC analysis as described above. The samples were stirred using the designed device at 200 and 1000 rpm.

Device's positions used for samples are C2, C7, D2 and D7 (Fig. 1).

## Data availability

The data supporting this publication and instructions for re-producing and constructing the stirring device are available at the SI and from the corresponding author upon request. Data is also available at https://zenodo.org/record/4118046.

## Code availability

Supplementary Dataset 3: ARDUINO firmware program code.

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

## Acknowledgements

O.M. is funded through the NWA StartImpuls. This work has been funded by the ERC (AdG 741774), the NWO (Vici grant 724.012.002) and the Dutch Ministry of Education, Culture and Science (Gravitation program 024.001.035). We thank Andreas Hussain for discussions.

## Author contributions

O.M. designed the device, with input from J.O. and O.V. J.O. designed and performed experiments, with input from O.M. O.V constructed the device. O.M. wrote the manuscript draft. O.M., J.O., O.V. and S.O. wrote, read and approved the manuscript.

## Competing interests

The authors declare no competing interests.
