## [Peer Review File · Communications Chemistry]

Reviewers' comments:

Reviewer #1 (Remarks to the Author):

This manuscript describes a stirring device which can operate inside an LC instrument's autosampler, in place of a 'traditional' vial tray. Continuous stirring of reactions within an autosampler allows direct sampling during reaction progress; with slower reactions, this allows effective time-resolved analysis. Importantly, the authors provide details on parts, construction, and validation protocols, to enable others to use the device they have developed. They provide data demonstrating time-resolved analysis of their lab's unique chemical system, which is sensitive to stirring.

I like this: both the content, and clear concise presentation. I can imagine myself and others using this device, and that publishing design information and validation protocols is where I feel the value of the manuscript lies (the thiol chemistry is already widely published).

I do have some concerns about how reproducible stirring experiments may be, though, and feel a number of improvements could be made to make the construction details and relevant protocols more accessible to colleagues wishing to use the device. As it stands, I would struggle to reproduce it, and would be 'starting from scratch' in attempts to validate it; that reduces the appeal to me. With this in mind, I've included some questions and comments below. [I appreciate that lab/office access may not be available to all in these difficult times; I have suggested including for evidence which I imagine the authors should have in hand, in one form or another]

Originality

- The Otto group thiol library system reported is widely reported.
- I have not come across any examples of similar devices in the peer-reviewed literature.
- I have made a cursory search, without finding anything in peer-reviewed literature.
- I have come across a number of examples, in person, and anecdotally, of similar devices (sometimes very similar, generally less accomplished). I also note colleagues also quoted unpublished examples in response to your Twitter post of a preprint of this manuscript.
- I would have hoped to see some recognition in the manuscript that such precedents exist, even if unpublished.
- BUT... unpublished/anecdotal precedents are an argument FOR publishing a manuscript such as this, not against it (they are not available to reproduce, and lack robust validation protocols demonstrates exactly why a manuscript like this is of value). The value (and originality) lies in others not having to "reinvent the wheel".

Validation: Validating the reproducibility of stirring across the device (since it is run from a central motor) is important. Any chemist who has run multiple large reactions clustered on one magnetic stirrer-hotplate will be aware that homogeneity of stirring is not a given (or stirring at all!). For synthetic chemists driving a reaction to completion, this is rarely a reproducibility issue; for a tool to study kinetics, though, robustly validating reproducible stirring is vital.

- We are not shown much of the data demonstrating validation: the authors report multiple repeats, but we are only see four repeats in Fig 3A (I presume this is four repeats from one well, showing considerable variance, although I found the labelling opaque). It would be helpful to see the full data (in the SI, at least) for all the wells, to understand to what extent reproducibility is really validated. Labelling of all the data figures should also be checked carefully (can the reader tell which line represents which experiment?).
- The authors mention that some positions were "visually confirmed" to rotate equally, while

others were different.

It will be particularly instructive to see the difference observed during between those positions during validation. (and how large that variance were, c.f. inter-run variance). Are you able to compare these?

- The authors attribute the variance between the different repeats that are shown (which seems non-negligible) to the stochastic nature of nucleation.

Perhaps this becomes more clear on viewing the full data, but 'by eye' it appears that initiation happens at the same time in Fig 3A, and it is the rate of progress of the reaction which varies. Perhaps the authors could provide some data on this to demonstrate that this results from the nature of the reaction, rather than poor reproducibility of stirring?

- "Visual inspection of positions A4, A5, C2, D2, C7, D7, F4 and F5 (Fig. 1) confirmed equal rotation of stirring bar,"

Does visually inspection confirmed mean "it looks the same" or that some visual measurement had been made (e.g. using video)? I imagine "eyeballing it" is quite a blunt instrument (this reviewer can struggle to tell the difference between 45 and 33.3 rpm by eye, without carefully reference, never mind hundreds of rpm). If you have no quantitative evidence, perhaps it would be more realistic to reduce claims of confirmation? Furthermore, if differences between some wells are so large that they are obvious by eye, then I imagine we should take it there to be 'invisible' differences, unless we have evidence otherwise (and the manuscript might reflect this) .

Have you tried filming it and counting? [I imagine that the slower rpm range should be accessible even to typical mobile phones; 200 rpm = 3.3 rps]. A protocol to conveniently quantify stirring with a microscope/video would be helpful for anyone trying to reproduce the setup.

Furthermore, if the authors are correct in suggesting that stochastic effects make their thiol reactions poorly reproducible, it is hard to consider them realistic as a practical quantitative validation for the device.

- You mention battery drain being considerable (65 % of the power in batteries drained over half a 150 h run). Could the variance in kinetics between runs instead result battery issues? Do you have any data on this?

[In my own work, I've fed DC power supplies into HPLC autosamplers; if you've not already validated the battery issue, I'd suggest that feeding in a stable power supply is probably less work!]

Hopefully the authors have data in hand which can allay these concerns. I'd suggest that if not, and they are not able to get it, then perhaps we might suggest some alterations to the text, reflecting the idea that we lack quantitative data on reproducibility, but that – in this poorly reproducible system – variance does not seem greater than the manual arrangements this device replaces.

Time-resolved study of "Otto Group" thiol library chemistry:

- The time-resolved data of the thiol system (e.g. in the video) is very pretty. It is a shame that this is not clearly represented in the manuscript. While not everyone is going to consider a 2D plot a thing of beauty, anyone who's ever done time-resolved studies by hand (there are a few of us), certainly will appreciate it.

- "Oxidation". Whiel the chemistry is not the focus here, it could be communicated a little more clearly. The oxidant in most "oxidation" reactions is not detailed, except for "pre-Oxidation" test. I presume this is air oxidation, but this should be made clear.

Can colleagues reproduce the device?

- Can the design files be included, instead of hosted independently? I tried to review these, and was denied access, which illustrates my concern over a vital part of the report being separated from the manuscript. [While the video is pretty, I'd suggest that construction details would be far more valuable, if server space is in short supply]

- It is not clear to me how this is put together, without knowing what it's mean to look like. I'd suggest that either the instructions are thoroughly illustrated, or – at least – that photos of the

internal parts of the stirrer, and any other details, are included.

[My apologies if this is included in the inaccessible externally-hosted material; it's not available to me]

- Including the Arduino code as a separate file that can be easily flashed onto the Arduino might also prove useful for first timers.

Can colleagues validate their own devices?

- While the validation experiments reported exploit the unique properties of the Otto lab system, this is not available to most users. It would be valuable to include a practical validation protocol which IS available to others, if this manuscript is to be of use to them.

[I hope that more thorough description of the "visual" validation mentioned earlier might serve here, if this is quantitative; if not, perhaps developing a simple protocol would be of value both to the authors and their readers]

Minor comments

- The manuscript could do with a little English language 'smoothing'.

- "Any experimental setup that requires continuous stirring can now be run inside the UPLC machine" may be overstating the case. Most "experimental setups" employed by my colleagues involve temperatures or other manipulations that are not facilitated here.

- "and consequently UPLC needle height was adjusted." In making instructions/details for reproduction clearer, I'd suggest including further warnings of the damage that can be done to an instrument by failing to note this!

- "facilitates high frequency and reproducibility of UPLC injections".

Isn't the frequency and reproducibility of LC injection is facilitated by the LC instrument (this device just manages not to interfere with that)? Perhaps it might be more valuable to conclude that this facilitates time-resolved measurements (e.g. for kinetics studies)?

- Perhaps the section title "UPLC sample preparation" is a little misleading: surely this is reaction setup?

- P2. "if the sampling quality is consistent" Do you mean sample volume (i.e. quantity)? Or something else? I note that you mention fibres falling to the bottom of vials: is that relevant to this?

Reviewer #2 (Remarks to the Author):

The manuscript reports on the fabrication of a 3D-printed device that has been designed in order to perform continuous stirring of samples while inside a UPLC system. The device was tested in a model transformation, a dithiol oxidation that leads to the formation of fibers, that is heavily influenced by the stirring conditions.

The device represents a useful application of the 3D printing technology, that allows to easily assemble equipment and devices, but in my personal opinion it is only an interesting technical device, that may be useful, in some specific applications. I am afraid I cannot recommend publication of the manuscript. Although the authors claim that "various tests were performed by following the kinetics of a dynamic combinatorial library...", indeed the study is limited to only one single transformation, and a limited number of data are reported. The paper is mainly directed to analytical chemists and should be published on a more specialized journal.

I also should add that references are largely incomplete; several reviews highlight the different areas of applications of 3D printing technologies and should be cited in the paper, at least to give a general overview of the activities in the field (either in analytical or organic chemistry, or processing or automation...)

Reviewer #3 (Remarks to the Author):

This manuscript sets out to describe a stirring device that can be placed within a UPLC in order to

continuously stir a set of reaction vials during UPLC sampling. The technique is multi-disciplinary in its design and construction but has an analytical chemistry technique at its core.

My main criticism of the paper is the title suggests that the paper will discuss the design and construction of the stirring device, but that very of this comes across in the paper itself. There is little discussion of the design, or design constraints, manufacture, type of 3D printing, CAD or stl files etc. The paper would benefit from this discussion and the inclusion of some of these elements in the SI. The 3D CAD files are apparently available online, but when I checked these are closed access. They should be available with this manuscript.

The chemistry itself is interesting and appears to support the authors' central thrust that stirring is important in some applications of UPLC.

Overall, this is an interesting manuscript, but the authors need to provide all the requisite information so it can be fully reviewed.

Manuscript title: "Automated stirring device for continuous stirring while sampling in liquid chromatography systems"

Reviewers' comments:

Reviewer #1 (Remarks to the Author):

This manuscript describes a stirring device which can operate inside an LC instrument's autosampler, in place of a 'traditional' vial tray. Continuous stirring of reactions within an autosampler allows direct sampling during reaction progress; with slower reactions, this allows effective time-resolved analysis. Importantly, the authors provide details on parts, construction, and validation protocols, to enable others to use the device they have developed. They provide data demonstrating time-resolved analysis of their lab's unique chemical system, which is sensitive to stirring.

I like this: both the content, and clear concise presentation. I can imagine myself and others using this device, and that publishing design information and validation protocols is where I feel the value of the manuscript lies (the thiol chemistry is already widely published).

We thank the reviewer for the nice words about our manuscript.

I do have some concerns about how reproducible stirring experiments may be, though, and feel a number of improvements could be made to make the construction details and relevant protocols more accessible to colleagues wishing to use the device. As it stands, I would struggle to reproduce it, and would be 'starting from scratch' in attempts to validate it; that reduces the appeal to me. With this in mind, I've included some questions and comments below.

[I appreciate that lab/office access may not be available to all in these difficult times; I have suggested including for evidence which I imagine the authors should have in hand, in one form or another]

We agree and have made improvements as suggested by the comments, addressing the reproducibility of the device and the validation of stirring. Please see our responses to the comments below.

Originality

- The Otto group thiol library system reported is widely reported.*
- I have not come across any examples of similar devices in the peer-reviewed literature.*
- I have made a cursory search, without finding anything in peer-reviewed literature.*
- I have come across a number of examples, in person, and anecdotally, of similar devices (sometimes very similar, generally less accomplished). I also note colleagues also quoted unpublished examples in in response to your Twitter post of a preprint of this manuscript.*

- I would have hoped to see some recognition in the manuscript that such precedents exist, even if unpublished.

We agree with these comments, which are part of the motivation we have to publish the device.

The Twitter post occurred after the manuscript has been submitted and therefore could not inform the version reviewed by the referees. Furthermore, we have later contacted the author of the tweet who said it was part of a grant application and therefore it is confidential and unpublished.

- BUT... unpublished/anecdotal precedents are an argument FOR publishing a manuscript such as this, not against it (they are not available to reproduce, and lack robust validation protocols demonstrates exactly why a manuscript like this is of value). The value (and originality) lies in others not having to "reinvent the wheel".

Validation: Validating the reproducibility of stirring across the device (since it is run from a central motor) is important. Any chemist who has run multiple large reactions clustered on one magnetic stirrer-hotplate will be aware that homogeneity of stirring is not a given (or stirring at all!). For synthetic chemists driving a reaction to completion, this is rarely a reproducibility issue; for a tool to study kinetics, though, robustly validating reproducible stirring is vital.

- We are not shown much of the data demonstrating validation: the authors report multiple repeats, but we are only see four repeats in Fig 3A (I presume this is four repeats from one well, showing considerable variance, although I found the labelling opaque). It would be helpful to see the full data (in the SI, at least) for all the wells, to understand to what extent reproducibility is really validated.

Labelling of all the data figures should also be checked carefully (can the reader tell which line represents which experiment?).

The validation shown in Figures 3 and 4 is based on four completely separate sets of experiments, which we now indicate in the manuscript. We have probed 4 different wells when measuring the dithiol system, and now also performed video analysis (see below) for 10 different wells.

We also improved the presentation of Figures 3 and 4 to make it less opaque, and we include in the SI a table with all the data plotted in those two figures.

- The authors mention that some positions were "visually confirmed" to rotate equally, while others were different. It will be particularly instructive to see the difference observed during between those positions during validation. (and how large that variance were, c.f. inter-run variance). Are you able to compare these?

Using the suggested video analysis, we now have quantified the stirring of the Teflon coated magnet in 10 different wells to find only a tiny difference – less than 1.5% - between the speed reported by the device. The device reports the speed of the disc with magnets by measuring via a magnetic sensor.

- The authors attribute the variance between the different repeats that are shown (which seems non-negligible) to the stochastic nature of nucleation.

Perhaps this becomes more clear on viewing the full data, but 'by eye' it appears that initiation happens at the same time in Fig 3A, and it is the rate of progress of the reaction which varies. Perhaps the authors could provide some data on this to demonstrate that this results from the nature of the reaction, rather than poor reproducibility of stirring?

With the quantification by the video analysis, it is now clearer that the variance between different repeats is indeed due to the stochastic nature of nucleation and not as a result of the stirring device.

- "Visual inspection of positions A4, A5, C2, D2, C7, D7, F4 and F5 (Fig. 1) confirmed equal rotation of stirring bar,"

Does visually inspection confirmed mean "it looks the same" or that some visual measurement had been made (e.g. using video)? I imagine "eyeballing it" is quite a blunt instrument (this reviewer can struggle to tell the difference between 45 and 33.3 rpm by eye, without carefully reference, never mind hundreds of rpm). If you have no quantitative evidence, perhaps it would be more realistic to reduce claims of confirmation? Furthermore, if differences between some wells are so large that they are obvious by eye, then I imagine we should take it there to be 'invisible' differences, unless we have evidence otherwise (and the manuscript might reflect this) . Have you tried filming it and counting? [I imagine that the slower rpm range should be accessible even to typical mobile phones; 200 rpm = 3.3 rps]. A protocol to conveniently quantify stirring with a microscope/video would be helpful for anyone trying to reproduce the setup.

We find the idea of video analysis a very good one, that would not only help to further validate the device but also is more accessible to potential users than a specific chemical system.

We have improved the sample holder design to make visual inspection easier (and include the design file), and have performed a video analysis. The video analysis confirmed that the rotations in the wells surrounding the central motor are indeed as they should (albeit at about 1% lesser speed) for 200 and 1000 rpm.

Furthermore, if the authors are correct in suggesting that stochastic effects make their thiol reactions poorly reproducible, it is hard to consider them realistic as a practical quantitative validation for the device.

We agree and believe that the video analysis provides a practical quantitative validation for the device. The results are reported in the text and the supporting information includes the data and the videos.

- You mention battery drain being considerable (65 % of the power in batteries drained over half a 150 h run). Could the variance in kinetics between runs instead result battery issues? Do you have any data on this?

[In my own work, I've fed DC power supplies into HPLC autosamplers;

if you've not already validated the battery issue, I'd suggest that feeding in a stable power supply is probably less work!]

This cannot be the case in our device, because it is managed by an Arduino microcontroller and has internal feedback on the live rotation speed of the plate with magnets (via a magnetic sensor). Because of this, as long as the battery level is sufficient for the operation of the microcontroller the rotations will continue regularly.

In a marginal case, when the battery is close the complete drainage (below 5% battery level), there might be some variance resulting from battery issues, but such a case will be very brief as the battery will quickly die. Moreover, the user always has indication of the battery level on the screen.

We have also updated the firmware code to report to the user if there were issues with stirring – if the measured live stirring speed is different than the user input speed, and included this in the text.

Hopefully the authors have data in hand which can allay these concerns. I'd suggest that if not, and they are not able to get it, then perhaps we might suggest some alterations to the text, reflecting the idea that we lack quantitative data on reproducibility, but that – in this poorly reproducible system – variance does not seem greater than the manual arrangements this device replaces.

Time-resolved study of "Otto Group" thiol library chemistry:

- The time-resolved data of the thiol system (e.g. in the video) is very pretty. It is a shame that this is not clearly represented in the manuscript. While not everyone is going to consider a 2D plot a thing of beauty, anyone who's ever done time-resolved studies by hand (there are a few of us), certainly will appreciate it.

Thank you.

We have added a plot with UPLC chromatograms to the SI.

- "Oxidation". While the chemistry is not the focus here, it could be communicated a little more clearly. The oxidant in most "oxidation" reactions is not detailed, except for "pre-Oxidation" test. I presume this is air oxidation, but this should be made clear.

You are correct, and we have indicated this in the text.

Can colleagues reproduce the device?

- Can the design files be included, instead of hosted independently? I tried to review these, and was denied access, which illustrates my concern over a vital part of the report being separated from the manuscript. [While the video is pretty, I'd suggest that construction details would be far more valuable, if server space is in short supply]

We agree and apologize for the negligence. The design files are included now.

- It is not clear to me how this is put together, without knowing what it's mean to look like.

I'd suggest that either the instructions are thoroughly illustrated,

or - at least - that photos of the internal parts of the stirrer, and any other details, are included.

[My apologies if this is included in the inaccessible externally-hosted material; it's not available to me]

We agree and have added photos of the individual parts to the SI text.

- Including the Arduino code as a separate file that can be easily flashed onto the Arduino might also prove useful for first timers.

We agree and include the Arduino code as a separate file. The updated version now also records and reports errors as we discussed above and in text.

Can colleagues validate their own devices?

- While the validation experiments reported exploit the unique properties of the Otto lab system, this is not available to most users. It would be valuable to include a practical validation protocol which IS available to others, if this manuscript is to be of use to them. [I hope that more thorough description of the "visual" validation mentioned earlier might serve here, if this is quantitative; if not, perhaps developing a simple protocol would be of value both to the authors and their readers]

We agree and have included the design files of the parts (included the version of the main holder which allows for better visual inspection, as we discussed above), and also included snapshots of them in the SI text.

Minor comments

- The manuscript could do with a little English language 'smoothing'.

- "Any experimental setup that requires continuous stirring can now be run inside the UPLC machine" may be overstating the case. Most "experimental setups" employed by my colleagues involve temperatures or other manipulations that are not facilitated here.

We have updated the text accordingly.

- "and consequently UPLC needle height was adjusted." In making instructions/details for reproduction clearer, I'd suggest including further warnings of the damage that can be done to an instrument by failing to note this!

We have added an additional note about this in the SI as well.

- "facilitates high frequency and reproducibility of UPLC injections".

Isn't the frequency and reproducibility of LC injection is facilitated by the LC instrument (this device just manages not to interfere with that)? Perhaps it might be more valuable to conclude that this facilitates time-resolved measurements (e.g. for kinetics studies)?

We agree and have modified the text accordingly.

- Perhaps the section title "UPLC sample preparation" is a little misleading: surely this is reaction setup?

We agree and have modified the text accordingly.

- P2. "if the sampling quality is consistent" Do you mean sample volume (i.e. quantity)? Or something else? I note that you mention fibres falling to the bottom of vials: is that relevant to this?

The "fibers falling to the bottom" is indeed relevant. If a sample is not homogeneously stirred then fibers precipitate and are therefore when the UPLC needle is sampling the fibers are under-represented, which results in loss of peak area. We have added text to clarify that.

Reviewer #2 (Remarks to the Author):

The manuscript reports on the fabrication of a 3D-printed device that has been designed in order to perform continuous stirring of samples while inside a UPLC system. The device was tested in a model transformation, a dithiol oxidation that leads to the formation of fibers, that is heavily influenced by the stirring conditions.

The device represents a useful application of the 3D printing technology, that allows to easily assemble equipment and devices, but in my personal opinion it is only an interesting technical device, that may be useful, in some specific applications. I am afraid I cannot recommend publication of the manuscript. Although the authors claim that "various tests were performed by following the kinetics of a dynamic combinatorial library...", indeed the study is limited to only one single transformation, and a limited number of data are reported. The paper is mainly directed to analytical chemists and should be published on a more specialized journal.

I also should add that references are largely incomplete; several reviews highlight the different areas of applications of 3D printing technologies and should be cited in the paper, at least to give a general overview of the activities in the field (either in analytical or organic chemistry, or processing or automation...)

We thank you for your comments.

We believe that the device, although being technical in nature as we and all the referees agree, may be relevant beyond the specific application in our laboratory as stirring and LC are widely used across different fields, and as such deserves being published. The relevance of the device is in bringing those features together which to the best of our knowledge was never reported before, as Reviewer #1 highlighted.

We have added several reviews highlighting the different areas of applications of 3D printing.

Following suggestions by reviewer #1, we have performed a video analysis that is a more accessible quantitative way for a potential user to verify the device.

Reviewer #3 (Remarks to the Author):

This manuscript sets out to describe a stirring device that can be placed within a UPLC in order to continuously stir a set of reaction vials during UPLC sampling. The technique is multi-disciplinary in its design and construction but has an analytical chemistry technique at its core.

We Thank the reviewer for the nice words about our manuscript.

My main criticism of the paper is the title suggests that the paper will discuss the design and construction of the stirring device, but that very of this comes across in the paper itself. There is little discussion of the design, or design constraints, manufacture, type of 3D printing, CAD or stl files etc. The paper would benefit from this discussion and the inclusion of some of these elements in the SI.

We have added details about the 3D printing to the text and commented on the open nature of the design.

The 3D CAD files are apparently available online, but when I checked these are closed access. They should be available with this manuscript.

We apologize and have included the design files in the revised manuscript, and also included images of them in the SI to make the manuscript and reproduction more accessible.

The chemistry itself is interesting and appears to support the authors' central thrust that stirring is important in some applications of UPLC.

Overall, this is an interesting manuscript, but the authors need to provide all the requisite information so it can be fully reviewed.

Thank you.

REVIEWERS' COMMENTS:

Reviewer #1 (Remarks to the Author):

This is much improved, and I look forward to seeing it published. As a chemist who "tinkers" with electronics, but is not an electronics expert, I believe that now we have a reasonable level of detail to enable colleagues to reproduce the device.

Below I include a couple of comments/thoughts, but do not see them as a barrier to publication, and leave it in the hands of authors/editors:

- The manuscript claims to print using PLA. In SI, there is talk about printing in ABS, with different settings on some, and make no mention of conditions on others. Is this "using PLA & these settings, unless otherwise stated"? If so, perhaps it's best to state that clearly.
- I would have hoped to see some recognition in the manuscript that this is not a new idea (though putting a reproducible procedure accessible to a chemist into the research literature IS new). I appreciate that the Twitter post I mentioned was made after submission. I'll leave this to the authors' ethical consideration.
- Were the revolutions observed in the video counter 'by hand', or by some automated process?
- I believe what I wrote last time answers Reviewer #2's concerns.

Reviewer #3 (Remarks to the Author):

The authors have addressed the points in the original referees comments: they have added CAD files, as well as providing video of the system in action. The paper can now be accepted for publication.

REVIEWERS' COMMENTS:

Reviewer #1 (Remarks to the Author):

This is much improved, and I look forward to seeing it published. As a chemist who "tinkers" with electronics, but is not an electronics expert, I believe that now we have a reasonable level of detail to enable colleagues to reproduce the device.

Below I include a couple of comments/thoughts, but do not see them as a barrier to publication, and leave it in the hands of authors/editors:
We thank the reviewer for reviewing the manuscript.

- The manuscript claims to print using PLA. In SI, there is talk about printing in ABS, with different settings on some, and make no mention of conditions on others. Is this "using PLA & these settings, unless otherwise stated"? If so, perhaps it's best to state that clearly.

We agree and have updated the text to indicate that.

- I would have hoped to see some recognition in the manuscript that this is not a new idea (though putting a reproducible procedure accessible to a chemist into the research literature IS new). I appreciate that the Twitter post I mentioned was made after submission. I'll leave this to the authors' ethical consideration.

We respectfully disagree as the tweet was done only after submission and therefore could not, and has not, informed our work or manuscript.

- Were the revolutions observed in the video counter 'by hand', or by some automated process?

Yes, the revolutions were observed 'by hand', by manually counting each video. This is indicated in the legend of Table S2 (previously numbered Table S1).

- I believe what I wrote last time answers Reviewer #2's concerns.

We agree.

Reviewer #3 (Remarks to the Author):

The authors have addressed the points in the original referees comments: they have added CAD files, as well as providing video of the system in action. The paper can now be accepted for publication.

We thank the reviewer for reviewing the manuscript.